# Learning To Estimate Search Progress Using Sequence Of States

**Matan Sudry, Erez Karpas**

Technion — Israel Institute of Technology, Haifa, Israel

matansudry@campus.technion.ac.il, karpase@technion.ac.il

## Abstract

Many problems of interest can be solved using heuristic search algorithms. When solving a heuristic search problem, we are often interested in estimating search progress, that is, how much longer until we have a solution. Previous work on search progress estimation derived formulas based on some relevant features that can be observed from the behavior of the search algorithm. In this paper, rather than manually deriving such formulas we leverage machine learning to automatically learn more accurate search progress predictors. We train a Long Short-Term Memory (LSTM) network, which takes as input sequences of states expanded by the search algorithm, and predicts how far along the search we are. Importantly, our approach still treats the search algorithm as a black box, and does not look into the contents of search states. An empirical evaluation shows our technique outperforms previous search progress estimation techniques.

## Introduction

Many interesting and challenging real-world problems can be defined as search problems, where the objective is to find a path in a state space (which is often implicitly defined by a black-box successor generator) from some given initial state to a goal state. Often, we can use a heuristic function which estimates the distance from any given state to the goal to better guide this search. Many heuristic search algorithms have been developed to solve such problems, the most well-known of which are probably $A^*$ (Hart, Nilsson, and Raphael 1968) and Greedy best-first search (GBFS) (Doran and Michie 1966).

Search algorithms can have unpredictable behavior, as we typically do not know the topology of the search space (if we did, perhaps we would not need to use a search algorithm). While there are some results when the search problem is described symbolically (Hoffmann 2011), these results do not apply for black-box search problem. Other work has addressed estimating search effort in terms of number of expanded nodes (Lelis, Stern, and Sturtevant 2014; Belov et al. 2017) or the cost of an optimal solution (Lelis et al. 2014) – we discuss these and others in more detail later. Importantly, these approaches are all *offline* – they give their predictions before the search starts.

We are interested in predicting remaining search effort *online*, that is, in estimating how much progress the search

algorithm has already made. This problem was first introduced by Thayer, Stern, and Lelis (Thayer, Stern, and Lelis 2012), who also derived some formulas for estimating search progress based on relatively simple search features, which we explain in more detail later.

Search progress estimation is a useful tool not just for letting a user know whether they have time to make a cup of coffee before search completes, but also in making decisions about the search. For example, remaining search time estimates are used in situated temporal planning (Cashmore et al. 2018) to prioritize search nodes which have a higher probability to lead to a timely solution. Such estimates can also be used to decide when to perform a restart in a Branch-and-Bound search (Anderson et al. 2019) for solving a Mixed Integer Program.

In this paper, we leverage recent developments in machine learning to automatically learn more complex search predictors based on a given training set. Specifically, we view the search algorithm as a black-box that emits a sequence of states (we also treat the node which the search algorithm expands as a black box), and train a Long Short-Term Memory (LSTM) network (Hochreiter 1998) to predict remaining search effort.

We conducted an empirical evaluation on a large set of planning domains, comparing our technique to previous work (Thayer, Stern, and Lelis 2012). The results indicate that our technique achieves state-of-the-art performance in search progress estimation.

## Background

In this section we review some of the necessary background on LSTM networks and heuristic search.

### LSTM Networks

Long Short-Term Memory (LSTM) networks (Hochreiter and Schmidhuber 1997) are a specific type of Recurrent Neural Network (Medsker and Jain 2001) – a deep learning approach which is especially suitable for learning from sequences. LSTMs were first introduced in order to solve the vanishing gradient problem in conventional Recurrent Neural Networks (Hochreiter 1998), and have been extensively used in natural language processing, e.g., (Sutskever, Vinyals, and Le 2014; Gers, Schmidhuber, and Cummins 2000). We now briefly review how LSTM networks are built,

but refer the interested reader to a detailed survey (Yu et al. 2019) for more details.

An LSTM network consists of several LSTM cells, arranged in sequence. Each LSTM cell has a hidden state, and 3 gates: input, forget and output, that manage the reading, writing and memory update, respectively. The cell learns gate weights and uses them to change the input and the hidden state based on the previous input in the sequence.

The cell processes input $x_t$ together with the hidden state $h_{t-1}$ (which is the result of processing the previous input) by first passing them to the forget gate, which has a Sigmoid activation function. The output of the forget gate is a number between 0 and 1 for each element in the cell state $C_t$ – the information we are passing to the next LSTM cell.

Next, the processing continues through the input gate, which has 2 activation functions, Sigmoid and tanh. The input is the same as the forget layer $x_t$ and $h_{t-1}$. The Sigmoid activation decides which values to update and the tanh activation creates the values themselves. The outputs from both of these are multiplied.

Finally, processing continues to the output gate that gets $x_t$ and $h_{t-1}$ as inputs and goes through a Sigmoid activation which decides what parts will be output. The output also goes through a tanh activation, which is multiplied by the output of the Sigmoid activation.

This is all captured by the following equations, which describe the operation of each LSTM cell:

- forget gate: $f_t = \sigma(W_f \cdot [h_{t-1}, x_t] + b_f)$
- input gate: $i_t = \sigma(W_i \cdot [h_{t-1}, x_t] + b_i)$
- new cell state value: $\widetilde{C_T} = tanh(W_c \cdot [h_{t-1}, x_t] + b_C)$
- update cell state: $C_T = f_t * C_{t-1} + i_t * \widetilde{C_T}$
- output gate: $o_t = \sigma(W_0[h_{t-1}, x_t] + b_0)$
- hidden state: $h_t = o_t * tanh(C_t)$

## Heuristic search

As previously mentioned, many problems of interest can be modeled as state-space search problems (Bonet and Geffner 2001). A state space is formally represented as a tuple $T = \langle S, S_0, S_G, A, f, c \rangle$, where:

- $S$ is a finite and non-empty set of states,
- $S_0 \in S$ is the initial state,
- $S_G \subseteq S$ is a non-empty set of goal states,
- $A(s) \subseteq A$ denotes the actions applicable in each state $s \in S$,
- $f : S \times A \to S$ is the state transition function, such that applying action $a$ in state $s$ leads to state $f(s, a)$, and
- $c(s, a)$ is the cost of performing action $a$ in state $s$

Uninformed search algorithms such as Breadth-First Search (BFS) can be used to solve small state-space search problems. However, to solve larger problems, we typically also use a heuristic which estimates the distance from any given node to the goal. Formally, a heuristic is a function mapping the states $S$ to some real number (or infinity to indicate a dead-end), that is $h : S \to \mathbb{R}_+ \cup \{\infty\}$.

Most search algorithms define some priority function $f(s)$ over the states. Typically, $f(s)$ is obtained from $h(s)$, the heuristic estimate of the distance to go from $s$, and $g(s)$, the cost of the best known path to $s$ so far. For example, A* (Hart, Nilsson, and Raphael 1968) expands the node with the lowest $f_{A^*}(s) = g(s) + h(s)$, while GBFS expands the node with the lowest $f_{GBFS}(s) = h(s)$. We will use both $f, g$ and $h$ depend on the search algorithm as features for each search node.

## Search progress estimation

We can now formally describe the problem we address in this paper – search progress estimation. Our definition here is adapted from previous work (Thayer, Stern, and Lelis 2012). Let $\mathbb{A}$ be a search algorithm, and $P = \langle T, h \rangle$ be a heuristic search problem, which consists of a search space $T$ and a heuristic function $h$. Denote by $E_\mathbb{A}(P)$ the number of nodes expanded by search algorithm $\mathbb{A}$ while attempting to solve a search problem $P$. Let $Rem_\mathbb{A}(P, E_\mathbb{A}(P))$ be the number of remaining nodes that are going to be expanded by $\mathbb{A}$ when solving $P$ after $\mathbb{A}$ has already expanded $E_\mathbb{A}(P)$ nodes, that is, $Rem_\mathbb{A}(P, E_\mathbb{A}(P)) = E_\mathbb{A}(P) - E_\mathbb{A}(P)$.

**Definition 1** (Search Progress). *The search progress of algorithm $\mathbb{A}$ solving problem $P$ after expending $E_\mathbb{A}(P)$ nodes is:* $Prog_\mathbb{A}(E_\mathbb{A}(P)) = \frac{E_\mathbb{A}(P)}{E_\mathbb{A}(P) + Rem_\mathbb{A}(E_\mathbb{A}(P))}$

As Definition 1 shows, search progress is a number between 0 and 1, which indicates what fraction of the total search effort to solve the problem we have already expended. For example if we expanded node number 1,000 out of 10,000 nodes that are going to be expanded in total, the search progress will be $1,000/10,000 = 0.1$. This allows comparing search progress between problems of different sizes on the same scale. The metric we use to evaluate the accuracy of search progress estimation (following (Thayer, Stern, and Lelis 2012)) is Mean Absolute Error (MAE): For each node we compute the absolute error between the predicted progress and the real progress. The accuracy of a predictor on a given problem is the mean of these absolute errors over all the states the search expanded.

## Related Work

Having described the necessary background, we can now con review related work in this field. Some previous work attempted to predict search effort in specific domains, e.g., predicting the number of nodes expanded by A* for the 15-puzzle (Breyer and Korf 2008). Other work used sampling to estimate search effort (Lelis et al. 2014; Belov et al. 2017; Hutter et al. 2014) or the cost of a solution (Lelis, Stern, and Sturtevant 2014) in a more general setting. However, as previously mentioned, these are all *offline* methods, which attempt to make a prediction before the search starts. We now review in more detail the only prior work we are aware of that addresses *online* search progress estimation (Thayer, Stern, and Lelis 2012). Several search progress estimators were proposed, which we now describe, and later use as baselines in our empirical evaluation:

**Velocity-Based Search Speed Estimator (VeSP)** The Velocity-Based Search Speed Estimator (VeSP) calculates the velocity during the search when $E_{\mathbb{A}}(P)$ denoted as the number of expanded nodes by the average velocity as:

$$V = \frac{h_0 - h_{\min}}{\text{Exp}}$$

It then uses the velocity $V$ and the current $h_{\min}$ to estimate the remaining search effort (number of expanded nodes)

$$SE_V = \frac{h_{\min}}{V}$$

Finally, it uses this estimate to predict search progress by using:

$$VeSP(\text{Exp}) = \frac{\text{Exp}}{\text{Exp} + SE_V}$$

**Vacillation-Based Search Speed Estimator (VaSP)** The Vacillation-Based Search Speed Estimator (VaSP) differs from VeSP by using node serial numbers to estimate the *expansion delay* (Dionne, Thayer, and Ruml 2011), that is, the average number of expansions between when a node is generated and when it is expanded, denoted $\overline{\Delta e}$.

The expansion delay is combined with $h_{\min}$ to predict remaining search effort as:

$$SE_e = \overline{\Delta e} \cdot h_{\min}$$

As in VeSP, the estimate of search progress is:

$$VaSP(\text{Exp}) = \frac{\text{Exp}}{\text{Exp} + SE_e}$$

**Path-Based Progress Estimator (PBP)** We first describe the Naive PBP (NPBP), which estimates search progress at a given node $n$ by looking at the ratio between the cost until the current node ($g(n)$) and the total estimated cost ($g(n) + h(n)$),

$$NPBP(Exp) = \frac{g(n)}{g(n) + h(n)}$$

The main problem with NPBP is that it depends on the current node, but in many cases there are other open nodes with higher progress. Thus the PBP estimator returns the maximum NPBP estimate among the nodes expanded far.

**Distribution-Based Progress Estimator (DBP)** The Distribution-Based Progress Estimator (DBP) estimates the search progress using data observed during search. It relies on counting how many nodes are expanded by the search for each value of $d(n)$ – an estimate of remaining plan *length*. We remark that in unit-cost domains $d(n) = h(n)$, and where action costs are non-uniform, $d(n)$ can usually be estimated alongside $h(n)$. Define $c[d_i]$ as the count of how many nodes had value $d(n) = d_i$ during search. DBP fits a second degree polynomial $\hat{c}$ to $c$, in order to estimate progress. The DBP estimate is then defined as:

$$Prog^*(\text{Exp}) = \frac{\text{Exp}}{\sum_{i=1}^{m} \hat{c}[d_i]}$$

where $m$ is the highest $d$-value with a count higher than 0.

## Learning to Predict Search Progress

We now describe our technique for search progress estimation, which is based on supervised machine learning. More precisely, given a training set of search nodes with their correct search progress, we attempt to predict search progress for new search nodes.

**Input Features:** We begin by defining the input features to our predictor. First, note that any expansion-based forward search algorithm works by expanding nodes in sequence. Thus, we can view the search algorithm as a black-box that outputs a sequence of nodes to expand. This motivates us to use a Recurrent Neural Network (as we discuss next), looking at the last $k$ nodes that were expanded. However, for each of these nodes, it is also important to know something about where it came from – the path from the initial node to that node. Thus, for each of these $k$ nodes, we also look at their ancestors $m$ layers up. In this paper, we used $k = 30$ and $m = 2$, resulting in collecting features from $k \cdot (m+1) = 90$ nodes, as illustrated in Figure 1.

For each of these $k \cdot (m+1)$ nodes, we collect the following features:

- $g(n)$ – the cost of the best known path to node $n$.

- $h(n)$ – the heuristic estimate of the cost from $n$ to the goal.

- $f(n)$ – the $f$-value for $n$ the search algorithm uses.

- $b(n)$ – the branching factor of $n$, that is, the number of successor node $n$ has.

- $N(n)$ – the serial number of $n$, that is, how many nodes were expanded before $n$. We remark that this number was previously used to measure the expansion delay (Dionne, Thayer, and Ruml 2011).

Additionally, we use some global features, which capture a snapshot of the current node of the search:

- $h_0 = h(S_0)$ – the heuristic value of the initial state. This value stays constant throughout the search, but is important for estimating the cost of the solution that will be returned.

- $h_{\min}$ – the minimal $h$-value we have seen so far among the expanded nodes.

- $Nh_{\min}$ – the number of nodes we expanded since the last time $h_{\min}$ was updated.

- $f_{\max}$ – the maximum $f$ value we have seen so far.

These features are used in the previous work (Thayer, Stern, and Lelis 2012), with the exception of $Nh_{\min}$, which is used to get some information about whether search seems to be stuck in a heuristic plateau (Asai and Fukunaga 2017). A heuristic plateau is an area of the search space where the heuristic assign similar values to all states, and thus heuristic guidance is not very useful. Search algorithms typically spend most of their time in heuristic plateaus, as otherwise, the heuristic would lead the search algorithm directly to the goal.

| ... | Grandparent (N-9) | Grandparent (N-8) | Grandparent (N-7) | Grandparent (N-6) | Grandparent (N-5) | Grandparent (N-4) | Grandparent (N-3) | Grandparent (N-2) | Grandparent (N-1) | Grandparent (N) |
|---|---|---|---|---|---|---|---|---|---|---|
| ... | Parent(N-9) | Parent(N-8) | Parent(N-7) | Parent(N-6) | Parent(N-5) | Parent(N-4) | Parent(N-3) | Parent(N-2) | Parent(N-1) | Parent(N) |
| ... | N-9 | N-8 | N-7 | N-6 | N-5 | N-4 | N-3 | N-2 | N-1 | N |

Figure 1: Illustration of the nodes used as input for prediction. $N$ is the node that was expanded last, $N - 1$ the node expanded before it, and so on. Parent() and Grandparent() are obtained by following the parent pointers for the indicated node.

We record the value of these features at the time each node was expanded, so we have $k$ values for each of these features. These input features can be seen as a matrix of real numbers, of dimension $5k(m + 1) + 4k$, which are used as the input (features) to our network. Note that for the first $k$ states, we use feature values of 0 for the undefined nodes (i.e., nodes with a negative serial number).

**Network Structure:** Our network[1] consists of 15 LSTM layers, followed by a fully connected layer which reduces the dimension by half (from 450 to 225), then a dropout layer with 50% dropout and Rectified Linear Units (ReLU) activations, and finally another fully connected layer which outputs our prediction – a single number. The parameters here (number of LSTM layers, dimensions, dropout rate, etc.) were determined by manual tuning over preliminary data. Automatically setting these parameters is future work.

**Training:** Given a planning problem, we first solve it using a planner running a given search algorithm and heuristic, and generate the true label (the correct search progress) for each node. During training, we sample 1,000 nodes uniformly at random from each problem in our dataset, in order to avoid bias to larger problems. We use the node serial number to obtain the true label of search progress – for node $n$ the true label is $N(n)/N(g_n)$, where $g_n$ is the goal node found by the search algorithm. We used Adam as an optimiser with a learning rate of 0.001 and a batch size of 1024, and Mean Squared Error (MSE) as the loss function. .

## Empirical Evaluation

Having described our technique and the previous work on search progress estimation, we can now compare them empirically. We now describe this empirical evaluation.

### Benchmarks

In order to compare search progress estimates on various problems with different heuristics and different search algorithms, we chose to use planning benchmarks from past International Planning Competitions, as this allows us to implement our technique only once. Specifically, we extended Pyperplan (Alkhazraji et al. 2020) under license GNU General Public License 3 (GPLv3) to output the features of the

---

[1] The code will be made publicly available upon publication of this paper.

nodes it expands during search. This data is then used for training and testing of search progress estimators, as we describe next.

Our dataset started with planning problems from 21 IPC domains – specifically, the benchmarks which are part of the Pyperplan repository. These domains have a total of 605 planning problem instances. For each instance, we ran 4 configurations of Pyperplan, choosing one of two search algorithms (A* or GBFS) with one of two heuristics (HFF (Hoffmann and Nebel 2001) or Lm-Cut (Helmert and Domshlak 2009)) with a time limit of 24 hours and a memory limit of 1,000,000 expanded nodes. From the solved instances we omitted the instances which were solved using less than 1,000 expanded nodes (see table 1 solved problems), as these are solved so quickly that search progress estimation is useless, and it would have made sampling 1,000 nodes from each instance difficult.

### Experimental setup

We compare our technique to the techniques introduced by previous work (Thayer, Stern, and Lelis 2012), which we described in detail above: VaSP, VeSP, PBP, and DBP. For VaSP, we used a moving average over the last 200 nodes to estimate expansion delay. For DBP we used NumPy to fit the polynomial to the data at hand.

All of our experiments were run on a server with 72 Intel Xeon E5-2695 CPUs (utilizing at most 18 processes in parallel (Tange et al. 2011) (available under GNU GENERAL PUBLIC LICENSE). The deep learning was implemented in PyTorch, and run on 2 Nvidia Tesla M60 GPUs. On this hardware the inference takes 0.05ms per node.

As our technique is based on learning from a training set, we examine three different training/test regimes:

**Other Domains (OD):** In this regime, when we test on problems from some domain, we train on all problems from all *other* domains. This simulates the setting where we need to estimate search progress on a problem from a completely unknown domain.

**Same Domain (SD):** In this regime, when we test on problems from some domain, we train only on problems from the same domain. To have some meaningful learning, we only used domains with more than 15 instances, leaving us only 3 domains. For each domain, we split its instances into the even- and odd-numbered problems, to obtain two sets of roughly the same size. We then trained a predictor on one set, and evaluated its performance on the other,

| | A$^*$/Lm-Cut | A$^*$/HFF | GBFS/Lm-Cut | GBFS/HFF |
|---|---|---|---|---|
| S | 121 | 119 | 89 | 204 |
| B | 11/19 | 10/18 | 13/15 | 12/20 |
| M | 23.4%/27.3% | 22.6%/24.3% | 20.6%/26.8% | 21.9%/26.0% |

Table 1: Experiment 1 summary on all configurations. S=Solved problems, B=Best accuracy, M=MAE ours/best predictor.

giving us 6 data points in total.

**Other Domains Tune Same (ODTS):** In this regime, we first train a predictor based on instances from the other domains. We then take this trained model, and fine tune it on one half of the instances from the target domain (split, as before, to even and odd numbered problems). During this fine tuning we run the same training algorithm, except that the learning rate is 0.0001 and the batch size is 2048. As fine-tuning requires less data than SD, we used domains with more than 6 instances – giving us 6 domains and 12 data points in total.

We measure the accuracy of each search progress estimate prediction (for a given state) by the absolute error between the prediction and the true progress (which we have, since we use only solved problems). The accuracy of a search progress estimator over a given problem is the average accuracy of its predictions over all the states expanded by the algorithm, that is the mean absolute error (MAE). The accuracy for a set of problems (all problems in a domain, or all problems overall) is the average accuracy across these problems (giving equal weight to each problem, regardless of the number of states it required expending), that is, the average mean absolute error. We also report the average of domain averages (that is, giving the same weight to each domain, regardless of how many problems it contains).

We remark here that we used MSE rather than MAE as the loss function during training, as MSE gives us a smoother function. Nevertheless, as we discuss next, our technique achieves start-of-the-art performance.

## Empirical results

We now describe the results for our empirical evaluation. First, Tables 2, 3, 4 and 5 compares our technique using the OD regime to the estimators proposed in previous work in all 4 configurations of heuristic and search algorithm. There are some missing values in the table, as some planner configurations do not solve any problem for this domain. We compare the previous work to OD, as it is the closest to these in terms of assumptions – it does not assume any knowledge about the problem it is used on.

To summarize these results more concisely, Table 1 shows in how many domains OD was the best predictor, and the overall accuracy of our OD predictor vs. the best predictor from the previous work. Overall, OD was the most accurate, with better average mean absolute error compared to the best predictor from previous work, VeSP/VaSP. The results when averaging over individual problems rather than domains are very similar.

| Domain | VaSP | VeSP | PBP | DBP | OD |
|---|---|---|---|---|---|
| Airport(5) | 29.4(13.6) | 19.4(3.0) | **19.3(3.0)** | 17.2(4.0) | 25.0(1.1) |
| Blocks(13) | 26.5(4.5) | 26.0(6.8) | 31.6(6.5) | 45.3(1.9) | **23.5(1.8)** |
| Depot(3) | 26.2(2.0) | 27.8(5.3) | 30.9(2.8) | 38.1(2.8) | **23.5(0.8)** |
| Elevators(11) | 23.9(2.2) | 24.6(4.4) | 29.5(4.2) | 34.0(5.1) | **22.0(3.1)** |
| Freecell(1) | 22.1(0) | **20.6(0)** | 32.3(0) | 33.4(0) | 24.2(0) |
| Gripper(4) | 38.9(2.7) | **18.5(1.5)** | 21.6(0.6) | 20.7(2.9) | 23.0(5.0) |
| Logistics(5) | 39.1(3.0) | 24.7(2.0) | 25.3(2.1) | 29.5(7.5) | **20.9(3.0)** |
| Openstacks(5) | **21.5(1.2)** | 33.9(3.2) | 37.3(3.1) | 37.3(3.7) | 25.1(0.4) |
| Parcprinter(6) | 34.3(6.4) | 34.9(2.5) | 35.1(2.7) | 40.6(2.0) | **23.4(3.2)** |
| Pegsol(15) | **23.4(2.7)** | 28.0(3.7) | 38.9(2.9) | 40.7(1.5) | 23.8(3.2) |
| Psr-small(15) | 28.2(4.4) | 26.5(2.4) | 41.9(1.9) | 40.6(1.8) | **22.8(2.6)** |
| Rovers(3) | 24.8(3.5) | 32.1(1.9) | 34.0(1.3) | 39.3(0.8) | **23.5(1.4)** |
| Satellite(2) | 30.5(4.1) | 22.9(3.6) | **22.9(6.7)** | 32.9(11.6) | 24.8(1.0) |
| Scanalyzer(3) | 23.3(2.6) | 25.1(2.8) | 29.4(1.3) | 27.0(6.2) | **22.3(0.4)** |
| Sokoban(16) | **24.4(8.2)** | 26.5(6.4) | 38.9(5.6) | 38.1(6.5) | 24.5(1.8) |
| TPP(1) | 30.5(0) | 39.9(0) | 40.8(0) | 41.9(0) | **24.4(0)** |
| Transport(5) | **21.8(2.8)** | 23.4(2.1) | 30.2(5.4) | 33.9(4.9) | 22.6(3.6) |
| WW(5) | 46.6(2.7) | 38.7(3.5) | 38.6(3.5) | 43.2(7.6) | **22.9(3.6)** |
| Zenotravel(3) | 26.9(3.2) | 26.0(6.8) | 30.0(6.9) | 29.2(9.1) | **22.1(1.4)** |
| Avg-dom(19) | 28.5(6.8) | 27.3(6.0) | 31.9(6.6) | 34.9(7.5) | **23.4(1.1)** |
| Avg-prob(121) | 27.6(7.7) | 27.0(6.0) | 33.9(7.4) | 36.9(8.0) | **23.3(2.6)** |

Table 2: Average mean absolute error (in percent) and standard deviation in parentheses for A$^*$ with Lm-Cut

| Domain | VaSP | VeSP | PBP | DBP | OD |
|---|---|---|---|---|---|
| Airport(2) | **21.8(8.4)** | 27.8(0.2) | 26.7(0.9) | 33.4(5.6) | 25.0(0.2) |
| Blocks(10) | 25.1(6.8) | 21.9(3.4) | 33.4(3.7) | 22.0(4.5) | **20.9(4.2)** |
| Depot(3) | 30.2(2.4) | 21.2(5.6) | 36.2(3.2) | 20.0(5.2) | **19.0(3.2)** |
| Elevators(2) | 22.0(6.5) | 15.9(3.4) | 33.6(2.0) | 14.5(1.9) | **14.2(0)** |
| Freecell(3) | 19.9(6.4) | 21.8(7.0) | 20.9(6.3) | 23.5(2.6) | **17.7(3.6)** |
| Openstacks(16) | 26.4(3.6) | 42.6(3.5) | 45.2(2.6) | 27.4(2.4) | **18.7(4.4)** |
| Pegsol(4) | 37.4(1.9) | 28.2(1.9) | 41.5(1.1) | 36.6(2.3) | **20.3(5.2)** |
| Psr-small(9) | 29.4(6.9) | 22.2(4.7) | 40.7(3.4) | 31.6(6.2) | **20.7(3.3)** |
| Rovers(4) | 25.2(4.1) | 22.4(4.2) | 32.9(4.9) | **21.5(4.6)** | 22.6(8.8) |
| Satellite(2) | 21.5(3.2) | 28.4(4.0) | 33.7(2.6) | 21.1(6.3) | **17.5(3.8)** |
| Scanalyzer(1) | 24.5(0) | 26.0(0) | 36.4(0) | 25.8(0) | **24.0(0)** |
| Sokoban(19) | 40.5(5.7) | 26.0(6.2) | 41.7(6.2) | 32.4(8.4) | **24.9(0.7)** |
| TPP(7) | 22.0(7.6) | 35.4(7.6) | 42.1(3.6) | 25.9(7.3) | **18.2(3.4)** |
| Transport(6) | 33.9(4.9) | 24.3(3.4) | 42.2(3.9) | 25.8(6.3) | **19.1(2.7)** |
| WW(1) | 22.4(0) | 43.7(0) | 44.3(0) | 38.0(0) | **16.7(0)** |
| Avg-dom(15) | 26.8(6.2) | 28.5(7.8) | 39.3(6.8) | 27.6(6.6) | **20.6(3.1)** |
| Avg-prob(89) | 29.7(8.6) | 27.2(9.1) | 36.8(6.9) | 26.6(7.4) | **20.0(4.4)** |

Table 3: Average mean absolute error (in percent) and standard deviation in parentheses for GBFS with Lm-Cut

| Domain | VaSP | VeSP | PBP | DBP | OD |
|---|---|---|---|---|---|
| Airport(8) | 15.6(4.2) | **13.0(5.9)** | 31.3(4.2) | 30.7(15.5) | 22.4(4.9) |
| Blocks(7) | 29.7(0.6) | 27.5(2.1) | 38.6(1.0) | 36.0(0.5) | **22.4(1.5)** |
| Depot(1) | 23.8(0) | **17.7(0)** | 34.4(0) | 24.5(0) | 18.0(0) |
| Elevators(5) | 26.5(1.5) | **21.3(3.1)** | 38.1(1.9) | 37.6(6.9) | 24.2(0.8) |
| Freecell(2) | 29.0(3.2) | 25.6(10.1) | 37.3(7.5) | 39.0(3.2) | **23.2(3.8)** |
| Gripper(4) | 26.9(1.4) | 48.0(2.2) | 35.8(4.0) | 44.4(5.6) | **21.2(2.5)** |
| Logistics(9) | 25.9(2.8) | 23.0(2.1) | 38.5(2.8) | 33.7(8.9) | **21.7(4.5)** |
| Openstacks(5) | 26.0(2.7) | **16.4(1.0)** | 38.7(2.7) | 21.1(1.3) | 21.7(3.8) |
| Parcprinter(9) | **17.4(1.7)** | 20.1(1.6) | 32.7(1.8) | 31.6(2.6) | 23.3(1.3) |
| Pegsol(15) | 34.5(0.4) | 28.0(9.1) | 42.9(1.6) | 42.1(1.1) | **23.6(0.7)** |
| Psr-small(16) | 30.5(4.6) | 40.8(12.3) | 42.0(2.3) | 37.6(6.0) | **22.9(2.1)** |
| Satellite(2) | 25.0(2.9) | 25.8(2.8) | 33.4(5.4) | 25.9(2.6) | **23.1(0.5)** |
| Scanalyzer(4) | 29.8(1.2) | 24.8(3.9) | 37.2(2.9) | 47.8(3.1) | **23.6(3.9)** |
| Sokoban(16) | 33.2(4.9) | 25.0(4.7) | 41.4(4.8) | 36.7(3.4) | **23.5(3.2)** |
| TPP(1) | 27.5(0) | **20.3(0)** | 38.3(0) | 26.4(0) | 22.3(0) |
| Transport(6) | 25.2(1.4) | **20.9(5.9)** | 37.9(2.4) | 35.2(3.0) | 22.7(3.2) |
| WW(5) | 29.9(0.5) | 26.1(1.3) | 38.3(1.9) | 45.5(4.2) | **23.5(2.5)** |
| Zenotravel(4) | 28.0(2.5) | **21.4(4.0)** | 38.3(2.5) | 37.7(3.3) | 23.8(3.7) |
| Avg-dom(19) | 27.0(4.7) | 24.3(8.3) | 37.6(3.1) | 36.1(7.5) | **22.6(1.4)** |
| Avg-prob(119) | 28.3(5.9) | 26.2(10.4) | 38.9(4.4) | 37.0(8.1) | **22.8(2.3)** |

Table 4: Average mean absolute error (in percent) and standard deviation in parentheses for A$^*$ with HFF

| Domain | VaSP | VeSP | PBP | DBP | OD |
|---|---|---|---|---|---|
| Airport(6) | **21.0(4.6)** | 38.8(8.9) | 26.9(3.9) | 37.3(7.8) | 21.3(2.1) |
| Blocks(20) | 27.3(2.9) | 30.9(5.3) | 41.5(2.3) | 35.5(7.0) | **20.6(2.5)** |
| Depot(19) | 26.1(5.0) | **21.6(3.3)** | 39.1(3.6) | 22.0(6.0) | 23.0(3.5) |
| Elevators(27) | 26.7(3.0) | **23.3(5.9)** | 38.3(3.5) | 26.2(4.4) | 24.1(2.9) |
| Freecell(14) | 15.0(5.5) | **13.8(5.9)** | 28.2(4.7) | 18.8(9.2) | 17.4(3.5) |
| Gripper(4) | 28.7(1.0) | 50.0(0) | 36.9(3.3) | 47.5(3.3) | **21.7(3.6)** |
| Logistics(11) | 24.4(4.4) | **15.0(2.9)** | 38.0(2.3) | 18.3(6.5) | 22.2(3.4) |
| Miconic(5) | **20.8(0.9)** | 37.9(4.7) | 32.1(3.8) | 25.8(6.6) | 21.2(5.1) |
| Openstacks(6) | 21.4(1.8) | 25.2(1.9) | 34.5(2.7) | **19.6(2.5)** | 22.7(2.9) |
| Parcprinter(8) | 22.2(6.8) | 21.4(2.7) | 29.6(4.3) | 21.6(4.1) | **21.3(4.4)** |
| Pegsol(18) | 40.0(3.2) | 29.2(9.7) | 45.4(1.2) | 40.4(6.4) | **22.9(3.6)** |
| Psr-small(12) | 29.5(5.3) | 40.0(12.4) | 42.0(3.5) | 36.1(9.0) | **21.2(1.9)** |
| Rovers(4) | 27.6(3.9) | **19.0(3.5)** | 39.8(1.7) | 30.5(9.1) | 21.1(3.2) |
| Satellite(1) | 26.1(1.4) | 27.8(0) | 35.7(3.6) | 25.0(0) | **20.6(6.5)** |
| Scanalyzer(5) | 29.4(5.2) | 29.4(12.6) | 36.4(8.5) | 35.6(7.2) | **24.9(6.0)** |
| Sokoban(20) | 38.9(5.8) | 26.1(4.6) | 42.2(6.8) | 33.4(7.0) | **23.9(3.4)** |
| TPP(4) | 20.2(4.4) | 22.0(3.8) | 32.1(3.0) | 20.3(5.6) | **19.9(3.6)** |
| Transport(8) | 25.9(2.1) | 23.2(5.0) | 37.9(1.9) | 23.9(6.3) | **22.9(3.1)** |
| WW(8) | 26.1(6.0) | 23.3(6.5) | 36.2(5.3) | 36.7(12.8) | **20.7(4.1)** |
| Zenotravel(3) | 22.4(3.1) | 30.2(7.4) | 28.1(9.4) | 22.6(2.1) | 24.6(4.0) |
| Avg-dom(20) | 26.0(5.9) | 27.4(8.9) | 36.0(5.2) | 28.9(8.4) | **21.9(1.8)** |
| Avg-prob(204) | 27.6(8.0) | 26.8(10.0) | 37.7(6.5) | 29.6(10.2) | **22.0(3.7)** |

Table 5: Average mean absolute error (in percent) and standard deviation in parentheses for GBFS with HFF

| Domain | OD | SD | ODTS |
|---|---|---|---|
| Blocks | **23.5(1.8)** | - | 23.6(2.0) |
| Elevators | 22.0(3.1) | - | **21.3(3.3)** |
| Parcprinter | 23.5(3.3) | - | **23.4(3.3)** |
| Pegsol | 23.8(3.2) | 25.2(0.1) | **22.6(3.1)** |
| Psr-small | 22.8(2.6) | 20.8(3.8) | **20.3(4.5)** |
| Sokoban | 24.5(1.8) | 24.3(2.1) | **23.5(3.1)** |
| Avg-dom(3) | 23.7(0.9) | 23.4(2.3) | **22.1(1.4)** |
| Avg-prob(46) | 24.3(2.7) | 23.7(3.1) | **22.2(3.5)** |

Table 6: Average mean absolute error (in percent) and standard deviation in parentheses with A$^*$ and Lm-Cut using all 3 regimes

To get a better sense of the behavior of these estimators during search, Figure 2(a,b,c) shows plots of the absolute error during search, for 3 problems of different sizes from different domains. For each point on the $x$-axis, which indicates the true search progress, we plot the absolute error of each predictor at that state. As we can see, VaSP has a lot of noise. This is explained by the fact that every change in the value of $h_{\min}$ causes a major change in the estimation. With respect to the other estimators, our method had better accuracy until about the 60–70 percent point, and from there the prediction was least accurate and missed the end of the problem. The difference between our model and the baselines at the end of the search is not surprising, as the baselines depend more closely on $h$, and thus when $h$ is close to zero, the estimates predict we are close to end of search.

Having seen that our technique, in the OD regime, outperforms previous work, we now turn to evaluating the impact of more specific training data by comparing the OD regime to the SD and ODTS regimes. Recall that in the SD regime our network is trained only on instances from the same domain, while in ODTS it is trained on instances from all other domains, and then fine-tuned on the domain being evaluated. Also recall that these regimes can only be used on domains which have enough data (15 solved instances for SD, 6 for ODTS), and thus the comparison is on fewer domains.

Tables 6,7,8 and 9, shows the results for this evaluation.

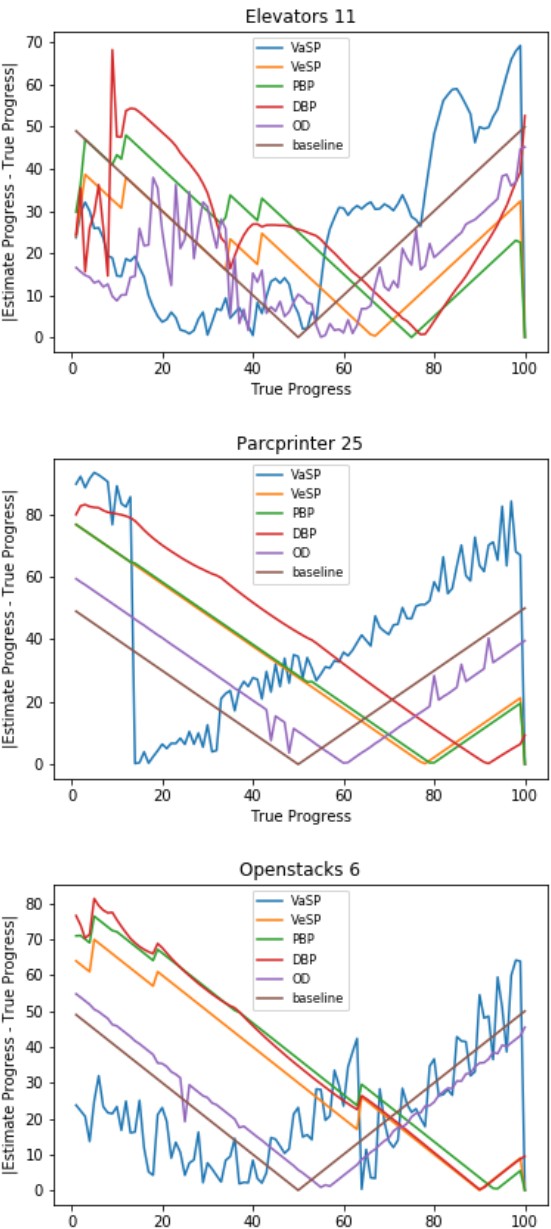

Figure 2: Absolute Error During Search in 3 Different Problems: (a) Elevators 11 (top, 4,500 nodes), (b) Parcprinter 25 (middle, 124,826 nodes), and (c) Openstacks 6 (bottom, 866,801 nodes)

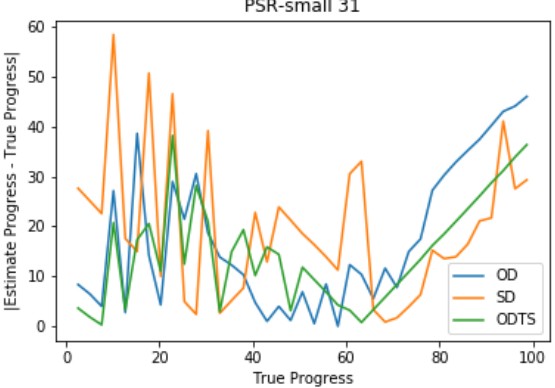

Figure 3: Absolute Error During Search in PSR-small 31 (3,950 nodes for all three training regimes)

| Domain | OD | SD | ODTS |
|---|---|---|---|
| Airport | **22.4(4.9)** | - | 22.5(5.3) |
| Blocks | **22.4(1.5)** | - | 22.4(1.4) |
| Logistics | **21.7(4.5)** | - | 22.7(5.7) |
| Parcprinter | 23.6(0.7) | **22.7(1.1)** | 23.0(0.8) |
| Pegsol | 22.9(2.1) | 20.8(3.1) | **19.9(3.6)** |
| Psr-small | 23.5(3.2) | 24.9(2.0) | **21.9(2.9)** |
| Sokoban | **22.7(3.2)** | - | 22.9(3.0) |
| Avg-dom(3) | 22.7(0.7) | 22.8(2.0) | **22.2(1.1)** |
| Avg-prob(47) | 22.8(2.9) | 23.3(2.8) | **21.7(3.5)** |

Table 7: Average mean absolute error (in percent) and standard deviation in parentheses with GBFS and Lm-Cut using all 3 regimes

| Domain | OD | SD | ODTS |
|---|---|---|---|
| Blocks | 20.9(4.2) | - | **19.5(4.5)** |
| Openstacks | 18.7(4.4) | 15.3(3.7) | **13.8(2.7)** |
| Psr-small | 20.7(3.3) | - | **19.5(4.3)** |
| Sokoban | 24.9(0.7) | 25.1(0.3) | **24.4(2.2)** |
| tpp | 18.2(3.4) | - | **17.4(2.9)** |
| transport | 19.1(2.7) | - | **20.3(3.0)** |
| Avg-dom(2) | 20.4(2.4) | 20.2(7.0) | **19.2(3.5)** |
| Avg-prob(35) | 20.6(4.1) | 21.0(5.5) | **19.4(5.0)** |

Table 8: Average mean absolute error (in percent) and standard deviation in parentheses with A* and HFF using all 3 regimes

| Domain | OD | SD | ODTS |
|---|---|---|---|
| Airport | 21.3(2.1) | - | **21.2(2.1)** |
| Blocks | 20.6(2.5) | **14.1(5.1)** | 15.7(3.7) |
| Depot | 23.0(3.5) | - | **22.3(3.6)** |
| Elevators | 24.1(2.9) | **21.7(4.1)** | 22.4(3.4) |
| Freecell | 17.4(3.5) | - | **16.4(4.5)** |
| Logistics | 22.2(3.4) | - | **21.1(3.8)** |
| Openstacks | 22.7(2.9) | - | **21.2(2.1)** |
| Parcprinter | 21.0(4.4) | - | **20.9(4.2)** |
| Pegsol | 22.9(3.6) | 23.2(3.2) | **22.6(2.8)** |
| Psr-small | **21.2(1.9)** | - | 21.9(2.7) |
| Sokoban | 23.9(3.4) | 25.0(2.8) | **23.5(2.9)** |
| transport | 22.9(3.1) | - | **22.2(2.6)** |
| Woodworking | 20.7(4.1) | - | **20.0(5.0)** |
| Avg-dom(4) | 21.8(1.8) | 21.2(4.5) | **20.9(2.3)** |
| Avg-prob(85) | 21.2(3.6) | 22.0(5.4) | **20.8(4.2)** |

Table 9: Average mean absolute error (in percent) and standard deviation in parentheses with GBFS and HFF using all 3 regimes

| test/train | GBFS/Lm-Cut | A*/Lm-Cut | GBFS/hff | A*/hff |
|---|---|---|---|---|
| GBFS/Lm-Cut | **19.8(3.1)** | 22.4(2.5) | 21.4(3.70) | 22.1(3.9) |
| A*/Lm-Cut | **23.6(1.8)** | 23.7(1.0) | 24.6(2.6) | 24.6(3.6) |
| GBFS/hff | 23.6(1.3) | 23.3(1.2) | **21.9(2.0)** | 22.4(2.2) |
| A*/hff | 23.2(2.4) | 23.1(2.7) | 23.0(1.7) | **22.6(1.5)** |

Table 10: Average mean absolute error (in percent) and standard deviation in parentheses on generalization evaluation

In the table there are missing values as well, since in some configuration the number of problems the planner solved was below the required number we define for this experiment. Interestingly, ODTS outperforms SD in most cases, showing that the training data from other domains is important. ODTS also outperforms OD in most domains as well as overall in each table, showing that domain-specific tuning does help. Figure 3 plots the absolute error of the 3 different regimes on a specific instance, PSR-small 31. As we can see, all 3 predictors started with unstable estimates, and got more and more accurate in the middle. Near the end all predictors become less and less accurate and missed the fact they are near the goal. Other than this, we can see that ODTS was generally more accurate, while OD and SD were each better than the other in different parts of the search.

## Generalization

We conclude our empirical evaluation by evaluating how our predictor handles changing the search algorithm, the heuristic, or both. In this experiment, we trained our predictor (using the OD regime) on one choice of search algorithm and heuristic, and then tested its accuracy on another choice. With 2 search algorithms and 2 heuristics, we have 4 combinations, and thus 16 settings to evaluate.

Table 10 shows the results of cross domain evaluation. As expected, using the same configuration for training and in test leads to the best performance in 3 out of 4 cases. The more interesting phenomenon is that when we change the algorithm and keep the heuristic the average MAE decreases by 0.86. When we change the heuristic and keep the algorithm, MAE decreases by 1.18, and changing both decreases

MAE by 1.31. Note that, for any configuration, training on the polar opposite (different search algorithm and heuristic) still yields better accuracy than the best predictor from previous work. This shows that our predictor can generalize to a different search algorithm and heuristic without suffering too severe a decrease in performance.

## Discussion and Future Work

In this paper we have showed a novel approach to search progress estimation using deep learning. We have shown that our approach outperforms previous state-of-the-art approaches, and that it benefits from having access to better training data. It is also interesting (and perhaps sad) to note that the previous search progress estimators (Thayer, Stern, and Lelis 2012), which were developed manually using extensive expertise in heuristic search and human creativity, are outperformed by a machine learning algorithm with very limited prior knowledge.

Besides being an interesting problem, search progress estimation can be used to make decisions about search. For example, as previously mentioned, estimates of remaining search time have been used in situated temporal planning (Cashmore et al. 2018) and in Branch-and-Bound search (Anderson et al. 2019). Other applications are in anytime search (Dionne, Thayer, and Ruml 2011) or metareasoning (Shperberg et al. 2019, 2020). However, using search progress estimates to make decisions about search has a problem with self-reference – the predictions affect search decisions, which affect search time, which then affect the true search progress. We intend to address this challenge in future work by attempting to learn a series of predictors which converge to a fixed point which exhibits good performance.

Finally, we remark that we employed fairly simple machine learning tools here. In future work we will examine whether we can improve performance even more by using different architectures of neural networks, or more informative features.

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
