# OpenReview forum: "Learning To Estimate Search Progress Using Sequence Of States"
_icaps-conference.org/ICAPS/2021/Workshop/HSDIP — HSDIP 2021_

### Official Review · AnonReviewer1 · 2021-05-26

**Confidence:** 4
**Overall Score:** Weak Accept

**Review:**

**Tittle:Learning To Estimate Search Progress Using Sequence Of States**

### Summary
The authors extract simple features from the last 30 nodes (and their 2 closest ancestors) expanded during the execution of a search algorithm. They use those features to train a recurrent neural network to predict how far the current search has already progressed. They compare themselves to other online search progress estimators and show that their approach produces better predictions (smaller absolute errors).

The authors present three different training regimes, training their estimator on instances of other domains, the same domain or training on instances of other domains and then fine tuning on few instances of the same domain. The approach with fine tuning performs best.

### Feedback
The prediction of the search progress is an interesting problem. In the extreme case, this could be used to probe how fast an algorithm is on a problem instances and to decide if the portfolio should continue with this algorithm or choose another one. I was surprised to see how uninformed all estimators were.

I think the approach and results from this paper are interesting and fit wonderfully the workshop. The results should be presented and discussed. At the same time, I think that especially from a stylistic point of view the paper has to be improved a lot (see minor issues below).

One of my biggest issue is with Figure 3. It shows how the mean absolute prediction error changes during the search for three selected instances. Firstly, I think it is really hard to understand from the plots what is happening. We directly see that all estimators show a V shape (although this shape is more pronounced for some estimators). How is this happening? This V shape means that the estimators approximately predict a constant value during any point of the search. The closer the progress of the search comes to this constant prediction, the smaller becomes the error. But any estimator which predicts a constant is uninformed and useless for this setting. **Add a baseline which predicts a constant value.** This helps the reader to interpret the plots correctly. As the estimators estimate mostly a constant, I though about the best constant and found out that if you always predict 0.5 you obtain the minimum mean error of *0.25*. Without any learning I give you a baseline with an mean absolute error of 0.25. All other baselines (see table 1) have a greater error. Can you please elaborate on the baseline techniques and my baseline. Have I understood something wrong, is there an error in the table, or what is the difference in the setup of the baseline techniques that justifies their bad performance? This reinforced my opinion that I would like to see the offline approaches as baselines too. At least you algorithm is better than my trivial baseline.

### Questions
- Why did you ignore offline approaches? I imagine that an offline approach predicts the number of expansions expected to solve the problem. If this is the case, it is very simple to transform an offline approach to an online one. Just add a counter which counts the number of expanded states and the progress is measures as $max(1,\frac{counter}{\text{predicted expansions}})$.
- Observing that many baseline estimators are almost constants, means that they are uninformed. I did not read up the relevant literature. Was this also observed in the original papers? If not, do you have an idea what is the important difference in your and their settings?
- A neural network with 15+ layers seems pretty large from the relatively few input features. You stated that the parameters were manually tuned. Have you tried using fewer layers? If yes, what where the results? If no, why not?

### Minor Issues
- *Background, LSTM, first paragraph*: This is a very long introduction to LSTM with few relevant information.
- *Background, LSTM, remaining paragraphs*: Following the textual description of an LSTM is very hard. Considering that most researchers at the workshops are no experts in the field of neural networks, I think this section has to be reworked. My suggestion: Add a figure visualizing an LSTM and shorten the text a lot (the schema contains the same information).
- *Background, Heuristic Search*:
	- $A$ is not introduced
	- Explicitly state that 0 is included in the co-domain of an heuristic: $h: S \mapsto \mathbb{R}^+_{0}$
	- "We will use $f$, $g$ and $h$ [...]". Which of the two previously defined $f$ functions do you use? $f_{A*}$, $f_{GBFS}$, or always the $f$ function of the search algorithm used (in the later text it is written that this will be the $f$ function of the search algorithm. Please clarify this here)?
- *Definition 1*: I would used $\mathbb{E}_A$ instead of $Exp + Rem_A(Exp)$ in the denominator, but that is your choice.
- *Figure 1*: The caption writes: "following the parent pointers". The plot denotes the parents as *Father* and *Grandfather*. Please unify this. I would simply use "Parent(N-X)" and "Parent(Parent(N-X))"
- *Figure 2*:
	- Update label: Distribution of *what* in the data? E.g. 'Distribution of the required expansions over all IPC instances.'
	- y-label: "(required) node expansions" or just "expansions"
	- Remove title of the plot. It's content is already in the caption of the figure.
	- x-label: What are the instance numbers? I am not aware of an instance numbering across domains. Change this to just "instances" or remove this completely.
	- Alternative/Suggestion: Change the plot to show a cumulative distribution: x-axis = number of required expansions, y-axis = probability, caption= something with cumulative distribution function showing how many expansions are required to solve the given instances
- *Empirical Evaluation, Benchmarks*: LMCUT -> LM-Cut
- *Empirical Evaluation, Empirical Results:* "In the other 8 domains, [...]" Skip those 5 words, they are more confusing than helpful.
- *Table 1:*
	- do **not** use camel case (Mean Absolute Error for Each ...)
	- "Domain/Estimator Combination" -> domain and estimator combination
	- Remove the "Avg problem" line (and the sentence speaking about it). It is biased. I was glad reading how you averaged within a domain and across all domains to get rid of biases. Just focus on "Avg domain".
	- I would like to see the standard deviations, but I also think the table is already hard to parse. If you can do something about it, I would be very interested in seeing your solution :)
- *Empirical Evaluation, Empirical Results, 3rd paragraph*:
	- This is about style. Figure 3 is just a plot, it does not have any intrinsic motivation. Thus, do not write that Figure 3 is evaluating something, but that YOU are evaluating it and Figure 3 shows the results.
	- State **in the text** the evaluated instances or at least that they are from different domains.
	- State how you selected the used instances for the evaluation.
- *Table 2:* Remove row "Avg problem"
- *Figure 3:*
	- no camel case in the caption
	- add to the caption something like "for all three training regimes"
- *Figure 4:* try to reuse the same colors as in Figure 3 for the same labels and if possible use new colors for new labels.
- *References*: In the Pyperplan references you state the DOI thrice and in another reference you state the DOI twice. This is ugly and redundant. Please update those references.

---

### Official Review · AnonReviewer2 · 2021-05-27
**The paper suggests to use LSTMs to estimate search progress and compares to hand-designed features**

**Confidence:** 4
**Overall Score:** Accept

**Review:**

The paper introduces a method that attempts to estimate the progress of a heuristic search algorithm by employing LSTM networks. Progress is measured in terms of the number of expanded states.

The authors use the heuristic value, branching factor, f and g values, and the serial number of states as input features of the net. This is extended with static features such as the minimal heuristic value seen so far. The net predicts the search progress.
The net architecture is based on some preliminary experiments.

The authors compare their approach to existing search progress measures that are not based on learning.

The paper is mostly well-written and fits the scope of HSDIP.

Comments:
- I think the evaluation could be done in a more systematic manner. You analyse that in different phases of the search, your approach performs differently. Why not split the search into "start, middle, end" and look at these three phases individually. An analysis similar to Table 1 for each of the phases would be interesting!
- At the end of the evaluation, you say that ODTS outperforms SD because it is training on multiple domains. Couldn't it be that this is just because of more data being available to the training?
- I disagree with the first part of your conclusion. The known estimators are based on a very simple analysis of the search progress to be as general as possible. Since the features used are similar to the input of your nets, it is not surprising that your approach performs better in terms of prediction accuracy. In the end, the net can learn characteristics of the heuristic, which the previous approaches do not take into account.

Questions:
- LMcut is probably somewhat "friendly" for your approach, since it behaves quite consistently. Do you think that the results will carry over to heuristics that are based on abstractions or landmarks?
- Similarly, the behaviour of GBFS can vary significantly across instances of a domain. Do you expect your models to be able to capture this?
- Your approach seems to generalize quite well across different domains. What about using different heuristics (potentially only within a single domain, so SD)?
- How long does the training take?
- What's the motivation for using Pyperplan and its benchmark set? Why do you not use all IPC benchmarks? It seems like a larger training set could be beneficial.

Minor:
- Intro:
-- "black-box search problem*s*"
-- "before *the* search starts"
-- "we view the search algorithm as *a black* box"
- Search Progress Estimation: "has already expand*ed* Exp nodes"
- Learning to Predict Search Progress:
-- "ancestors ** m layers up"
-- "number of successor* node*s*"
- Related work:
-- the indentation of the method descriptions looks a bit strange
-- VeSP: "Estimator (VeSP) estimator"
- What is "\^{c}"?
- expended => expanded (several times)
- Table 1 is too wide

---

### Author Response · Authors · 2021-05-30
**Review Response**

We would like to thank both reviewers for their useful feedback. We will address all minor comments in the final version, but we respond to the major comments and questions below. We also remark that since the submission we have new results with a new search algorithm (GBFS) and a new heuristic (hFF - relaxed plan length), and we have results showing transfer of training on one planner configuration (search algorithm + heuristic) and testing on another. We will include these results in the final version.

R1
Comments:
We are not fully sure how to split the evaluation into 3 different phases. Our analysis of the structure of the plots was qualitative and not something we know how to do exactly. Specifically it is not clear when to switch between phases.
Yes, ODTS definitely benefits from having access to more training data. We will make this clear in the final version.
The first part of the conclusion is our opinion, but we will make sure to qualify it.
Questions:
Lmcut is an inconsistent heuristic which works by generating landmarks, so we would argue that our results already work there. We also have new results with the relaxed plan (hFF) heuristic, which are very similar. We do not have results with abstraction based heuristics, but we assume they will also be similar.
Yes, we also have new results with GBFS, and they are also quite good.
We have new results that show our predictor can generalize across different heuristics (lmcut to hFF or vice versa), across search algorithms (A* to GBFS or vice versa), and across both, suffering a small loss in accuracy but remaining more accurate than the baseline predictors.
Training takes 500 seconds per domain, on average. We will include this in the paper.
We used pyperplan because it’s implemented in Python and easier to modify and interface with neural network packages. For benchmarks, we simply used the domains that ship with Pyperplan.

R2
You are right that predicting a constant 50% progress would give a mean absolute error of 25%, and the plot would look like a V, which will be center at 50%. Note that most other predictors form the general shape of a V, but it is not always centered at 50%, showing they are providing some signal. We also remark that, had we plotted the error (including the sign), it would be possible to see that sometimes the prediction is too high and sometimes too low - this is obscured by using absolute error, but is needed to make the plots legible. We also remark that Thayer et. al. (2012) included similar plots in their paper.

Comments:
Thank you for the suggestion for using offline estimators. We will try to do so in a future version of this paper.
The baseline predictors were originally evaluated on a different set of domain-specific search problems. Perhaps this is the reason for the difference in their performance.
15 layers were chosen during initial trials empirically. It is possible that this is not the optimal value for the final set of features and network configuration, but we believe the differences will be minor.

---

> ### Comment · AnonReviewer1 · 2021-05-31
> **Answer**
>
> Thank you for your answers.
>
> It does not matter where the other predictors are centred. As long as they predict almost constants, they are as informed as predicting 0.5, just with a worse mean absolute error. Therefore, I would like that you add this simple baseline in the paper.
>
> Just another thought of me, but do not feel pressured to use it. Instead of the current MAE plots which use only one instance, can you aggregate the MAE across all instances of a domain (same axes as now)? The mean MAE across all instances becomes the line (which is now already plotted) and the standard deviation is plotted as an area around the line. With one or two techniques that should work, but I do not know if the plot is still readable with your number of techniques :)

---

> ### Comment · AnonReviewer2 · 2021-06-01
> **Thank you for the response**
>
> Regarding the split evaluation: I was thinking of an evaluation similar to Table 1, where for each variant there are three entries, one for the error in first 33% of the search (in terms of actual expansions), one for the middle 33%, and another one for the last 33%. (that's just one option, there could be more meaningful splits)
> That way, you get a more fine-grained picture, where, for example, a bad performance at the beginning of the search does not lead to a bad overall performance. It could tell if method X is more accurate close to the initial state, but method B gets better towards the goal.
>
> Thanks for the other comments, in particular the summaries of the new results. That's exactly what I was interested in. Good to see, and somewhat surprising, that your approach generalizes so well.

---

### Decision · Program_Chairs · 2021-06-10

**Decision:**

Accept

**Comment:**

Both reviewers are positive about the paper and recommend acceptance. We ask the authors to address the comments raised by the reviewers in the camera-ready version.